# Prevalence of *Chlamydia trachomatis, Neisseria gonorrhoeae,* and *Mycoplasma genitalium* among Patients with Urogenital Symptoms in Istanbul

**DOI:** 10.3390/healthcare11070930

**Published:** 2023-03-23

**Authors:** Hayriye Kirkoyun Uysal, Muammer Osman Koksal, Kutay Sarsar, Mehmet Ilktac, Zeynep Isik, Deniz Bahar Akgun Karapinar, Mehmet Demirci, Betigul Ongen, Ahmet Buyukoren, Ates Kadioglu, Eray Yurtsever, Ali Agacfidan

**Affiliations:** 1Department of Medical Microbiology, Istanbul Faculty of Medicine, Istanbul University, Istanbul 34093, Turkey; 2Faculty of Pharmacy, Eastern Mediterranean University, Famagusta 99450, Cyprus; 3Department of Medical Microbiology, Faculty of Medicine, Kirklareli University, Kırklareli 39100, Turkey; 4Department of Obstetrics and Gynecology, Istanbul Faculty of Medicine, Istanbul University, Istanbul 34093, Turkey; 5Department of Urology, Istanbul Faculty of Medicine, Istanbul University, Istanbul 34093, Turkey; 6Department of Biostatistics, Istanbul Faculty of Medicine, Istanbul University, Istanbul 34093, Turkey

**Keywords:** chlamydia trachomatis, neisseria gonorrhoeae, mycoplasma genitalium, urogenital infections, sexually transmitted diseases

## Abstract

*Chlamydia trachomatis, Neisseria gonorrhoeae*, and *Mycoplasma genitalium* are the three most commonly reported sexually transmitted bacteria. The present study aimed to investigate the presence of *C. trachomatis, N. gonorrhoeae,* and *M. genitalium* in urogenital samples collected from 18–68-year-old Turkish patients who were admitted to the hospital with various urogenital symptoms. A total of 360 patients with symptoms of STD were included in the study. Following DNA extraction by QIAamp Mini Kit, the presence of *C. trachomatis, N. gonorrhoeae,* and *M. genitalium* were investigated using multiplex real-time PCR. Causative organisms were identified in 68 (18.9%) of 360 patients. *C. trachomatis, N. gonorrhoeae*, and *M. genitalium* were detected in 40 (11.1%), 14 (3.9%), and 28 (7.8%) of the patients, respectively. Patients 21–30 years of age represented more than one-third (37.8%) of positive patients. Of all patients, dual infections of *C. trachomatis–M. genitalium, N. gonorrhoeae–C. trachomatis, N. gonorrhoeae–M. genitalium,* and triple infection of *C. trachomatis–N. gonorrhoeae–M. genitalium* were determined in 1.6% (6/360), 1.3% (5/360), 0.2% (1/360), and 0.2% (1/360) of the patients, respectively. In CT-, NG-, and MG-positive patients, different STI agents were also found such as HIV, HBV, HPV, HSV2, T. pallidum, and T. vaginalis. In conclusion, among *C. trachomatis, N. gonorrhoeae,* and *M. genitalium*, CT was the most frequently detected bacterial cause of STDs in our hospital at Istanbul. Co-infections, which comprise more than one-fifth of the cases, should not be underestimated. Regular screening and following up of STD agents using multiplex real-time PCR-based diagnostic methods enabling the immediate detection of co-infections are essential for the treatment and primary prevention of STDs.

## 1. Introduction

Sexually transmitted bacteria, generally detected in sexually active individuals, can be responsible for various manifestations ranging from asymptomatic or spontaneously resolving urethritis and vaginitis to severe, life-threatening diseases. Although sexually transmitted bacterial diseases are unlikely to be fatal in the acute phase, they pose a significant public health threat, especially to high-risk individuals for whom early detection and immediate treatment are crucial [1,2].

*Chlamydia trachomatis* (CT) infections are the most frequently reported sexually transmitted bacterial infections worldwide. Because of the high prevalence of infection, the Centers for Disease Control and Prevention (CDC) recommends routine annual screening of sexually active women aged 25 years old or younger and men who have sex with men, irrespective of condom use. CT infections can be asymptomatic or present with a wide range of symptoms, including dysuria, pelvic pain, vaginal discharge, dysmenorrhea, painful sexual intercourse, and testicular or rectal pain depending on the body site affected a result of sexual behavior and route of exposure. CT infections should be suspected in women with cervicitis, salpingitis, infertility, or pelvic inflammatory disease (PID) and in men with urethritis and epididymitis. Symptoms of anorectal pain and anal discharge should not be overlooked among individuals who have engaged in anoreceptive intercourse. Women with PID complicated by tubo-ovarian abscesses may experience systemic symptoms such as fever and chills [3,4,5,6,7].

*Mycoplasma genitalium* (MG) is a sexually transmitted bacterium responsible for acute or chronic non-gonococcal urethritis among men and urethritis, cervicitis, or PID among women. Similar to CT, most infections caused by MG are asymptomatic. Patients with persistent PID, urethritis, or cervicitis should be tested for MG. Screening of MG and CT infections is crucial not only for diagnosing sexually transmitted diseases (STDs) in symptomatic patients to deliver optimal care but also for detecting asymptomatic individuals who are a reservoir of the infection to prevent the spread of the disease [8,9,10]. 

The etiological agent of gonorrhea disease, *Neisseria gonorrhoea* (NG), is the second most frequently reported cause of STDs worldwide. In men, NG is responsible for genital tract infections such as urethritis and epididymitis that present with dysuria, urethral discharge, scrotal pain, and rectal infections characterized by rectal bleeding, anal itching, and pain during bowel movement. A sore throat may also develop in individuals who have had oral sex. In women, although other body sites may also be included, the primary site of infection is the cervix, in which the bacterium infects cylindrical epithelial cells. Because it cannot infect squamous epithelial cells lining the vagina, NG does not cause vaginitis. Urogenital discharge, dysuria, and abdominal pain are the primary symptoms detected in symptomatic women infected with NG. Complications such as endometritis, salpingitis, pelvic peritonitis, tubo-ovarian abscess, and disseminated gonococcal infection can develop in untreated patients [11,12,13,14].

Apparent symptoms and physical examination findings can facilitate the rapid diagnosis of STDs. However, due to similar urogenital signs and symptoms shared by a wide range of microorganisms, including bacteria, viruses, fungi, and the parasite, *Trichomonas vaginalis*, microbiological diagnosis is crucial for STD management. A rapid and sensitive diagnostic technique is essential for accurate, early diagnosis and prompt treatment of STDs, improving prognosis, preventing complications, and preventing the spread of infection. Microscopical analysis, culture, and molecular methods such as nucleic acid amplification tests (NAATs) are the techniques used in diagnosing STDs. Because of their higher sensitivities compared to the others, NAATs such as polymerase chain reaction (PCR) have recently been the recommended diagnostic tests. In addition to their superior sensitivities, NAATs also enable simultaneous testing of a broad panel of STD-related pathogens, which is of great value considering the high incidence of co-infections that result from the acquisition of multiple STIs simultaneously or concurrently due to common risk-associated behaviors. Recently, laboratories have increasingly offered various combinatorial multiplex NAATs, including CT, NG, MG, and various other pathogens, for diagnosing STDs. Increased rates of co-infections have been reported together with the use of PCR-based tests because otherwise the detection would be expensive and time-consuming. NAATs are helpful not only for the diagnosis but for clarification of the epidemiology and burden of sexually transmitted diseases (STDs) and also for advancing the knowledge of infections which are caused mainly by pathogens such as mycoplasma species that are difficult to investigate using conventional methods [15,16,17]. The present study aimed to investigate the presence of CT, NG, and MG in the urogenital specimens collected from 18–68-year-old patients with urogenital symptoms using a multiplex real-time PCR test.

## 2. Materials and Methods

### 2.1. Subjects and Sample Collections

The study was carried out between July 2019 and July 2021 in the Departments of Obstetrics and Gynecology, Urology, and Medical Microbiology at Istanbul University, Istanbul Faculty of Medicine Hospital. The study was approved by the Ethics Committee of Istanbul Faculty of Medicine (approval number: 816988 and date of approval: 30 March 2022), and written informed consent from the patients was obtained. 

We planned this study as a block randomization-based study for a period of 2 years. In order to avoid a bias in collecting patient materials, we did not interfere with the clinical situation or the age of the patients. In order not to concentrate on only one gender (female), we ensured that gender was matched, and after the number of our patient group met the sampling power, we ended the material collection phase of the study in which the genders of men and women were equal. Data on other STI parameters such as HIV (anti-HIV1-2 ELISA and HIV RNA), HBV (HBsAg, anti-HBC IgM, and HBV DNA), HPV DNA, HSV1 and 2 DNA, *Trichomonas vaginalis*, and *Treponema pallidum* (VDRL and TPHA) for all participants included in our study were extracted from the hospital laboratory automation system and the laboratory record system of the microbiology department. 

A total of 360 urogenital specimens were collected from 180 female and 180 male patients with urogenital symptoms. Urethral, cervical, and vaginal samples were collected using dacron-tipped swabs. In addition, twenty milliliters of the first void urine or urethral swab samples were obtained from men. Swabs were transferred to the laboratory in a transport medium containing sucrose phosphate buffer. 

### 2.2. DNA Extraction and Multiplex Real-Time PCR

DNA was extracted from the specimens using the QIAamp Mini Kit (Qiagen, Hilden, Germany), and DNA was stored at −20 °C until further testing.

All specimens were screened for CT, NG, and MG by a commercially available triplex polymerase chain reaction (PCR) kit (FTD Urethritis Basic Kit, Fast Track Diagnostics, Luxembourg) according to the instructions of the manufacturer. The FTD urethritis basic kit was a commercially available triplex real-time PCR kit capable of simultaneous and quantitative detection of NG, CT, and MG. It contains different colored probes targeting the adhesin (*mgpB*) gene of *M. genitalium*, the cryptic plasmid ORF8 gene of *C. trachomatis,* and the *opaK* gene for the opacity protein of *N. gonorrhoeae.* In addition to target genes in real-time PCR reactions, the kit also uses a separate stained probe for murine cytomegalovirus (MCMV) as an internal control to determine the quality of DNA extraction [18]. 

A real-time PCR reaction was performed with 25 µL of the total volume, which comprises 12.5 µL of buffer, 1.5 µL of primer probe mix, 1.0 µL of enzyme mix, and 10.0 µL of extracted DNA. Cycling conditions of real-time PCR were adjusted on RotorGene-6000 (Qiagen, Germany) as follows: 50 °C for 15 min, 94 °C for 1 min, followed by 40 cycles of 94 °C for 8 s and 60 °C for 1 min. Specimens with a cycle threshold value of ≤33 were considered positive. Positive controls (containing different plasmids for detecting CT, NG, and MG) and negative controls (including the lysis buffer) were used for every real-time PCR run. Quantification analysis and the amplification curves and evaluations according to Ct values were performed automatically in the real-time PCR instrument software according to the manufacturer’s instructions.

### 2.3. Statistical Analysis

Data were analyzed using SPSS 26.0. Descriptive statistics assessed demographic characteristics. Bi-variate statistical analysis independent tests and Chi-square tests were used to examine the relationship between the demographic characteristics and main measures. A *p*-value of ˂0.05 was considered statistically significant.

## 3. Results

Of the 360 patients included in the study, 140 (38.9%), 138 (38.3%), 42 (11.7%), and 30 (8.3%) were diagnosed with vaginitis, urethritis, cervicitis, and infertility, respectively. The remaining 12 patients had salpingitis, epididymitis, or PID. Of all patients, 68 (18.9%) were found to be positive for at least one of the three pathogens. CT, MG, and NG were identified in 40 (11.1%), 28 (7.8%), and 14 (3.9%) patients, respectively. Patients who were 21–30 years of age represented the majority (37.8%) of patients infected with at least one pathogen, followed by those who were 31–40, 41–50, ≥51, and ≤21 years of age. The correlation between age and the presence of infection was found to be statistically significant only for MG (Table 1).

CT (11.1%) was the most frequently detected pathogen in the samples collected from women, followed by MG (7.8%) and NG (3.3%). The primary pathogen in the urogenital samples of men was CT (11.1%), followed by MG (7.8%), and NG (4.4%). No statistically significant difference was detected between the gender and the presence of infection for any pathogens (Table 1).

CT, MG, and NG positivities were 7.9% (11/140), 7.9% (11/140), and 2.1% (2/140) among women with vaginitis, respectively. CT (21.4%) was the predominant pathogen in 42 patients with cervicitis, followed by NG and MG (7.1% each). CT was detected in 18 (13%), MG in 14 (10.1%), and NG in 8 (5.8%) of 138 patients with urethritis. Among 30 patients with infertility, 2 (6.6%) were CT-positive. No pathogen was detected in 12 patients with salpingitis, epididymitis, or PID. The difference between the clinical presentation and the detection of CT, NG, and MG was not found to be statistically significant. Overall positivities of CT, NG, and MG were higher in patients living in urban regions (24.7%) than those living in suburban (7.5%) regions. However, the difference was not statistically significant (Table 1).

Single infections were detected in 55 (80.9%) and co-infections in 13 (19.1%) of 68 patients who were STD-positive. Single infections of CT, MG, and NG were identified in 41.2% (28/68), 29.4% (20/68), and 10.3% (7/68) of the patients with STIs, respectively.

Dual co-infections of CT with MG, NG with CT, and NG with MG were detected in six, five, and one patient(s), respectively. One of the patients with NG–CT coinfection was an individual living with HIV. In one patient, triple co-infection of CT–NG–MG was determined (Table 2 and Figure 1).

One of the three patients infected with CT and one of the seven infected with MG were individuals living with HIV. One of the patients with NG–CT coinfection was an individual living with HIV. In three patients, the presence of *Treponema pallidum* was also detected. Table 3 shows the co-infections with different STI parameters.

After calculation with a Rogan–Gladen estimator, the true prevalence was calculated, and Table 4 was created.

## 4. Discussion

STDs are among the most critical public health problems in developed and developing countries because of acute or chronic diseases such as urethritis, epididymitis, vaginitis, cervicitis, and reproductive health problems that they cause. NG, CT, MG, *Treponema pallidum*, and *Ureaplasma urealyticum* are among the most frequently encountered bacterial agents worldwide, as well as in Turkey. In addition, because sexual contact is the primary transmission mode, the incidence of STDs is the highest among sexually active men and women aged 15 to 49 [7].

Conventional laboratory methods such as Gram staining and culture are limited to diagnosing NG. Moreover, these methods are generally insensitive, especially if the patient is consuming an antibacterial drug, the specimen is transported to the laboratory under inappropriate conditions, or the transport of the specimen is delayed. In addition, microscopical analysis is inherently subjective, and the culture technique is laborious and has long turnaround times of approximately 48–72 h. On the other hand, molecular-based assays that investigate the target genomic sequences of various pathogens responsible for STDs directly from urogenital specimens, thus bypassing the requirement of culture, can simultaneously investigate a wide variety of causative agents in a multiplex manner, allowing the detection of co-infections, decreasing the turnaround time, and contributing to immediate and accurate treatment. Molecular-based multiplex PCR methods are also flexible that can be modified to cover the locally most frequently detected bacteria in a single test. Pathogens responsible for STDs should be investigated using highly reliable methods to accurately determine their prevalence [19].

As a result of the study carried out by Barrientos-Durán et al. [20], who compared the diagnostic accuracies of two commercially available NAAT-based kits [Aptima Assays (Hologic) and CoBAS^®^ 6800 system (Roche)] for detecting CT, NG, MG, and *T. vaginalis* from the clinical samples, the overall sensitivity and specificity were reported as 98.9% and 100% for Aptima assays and 100% and 96.67% for CoBAS, respectively. Although the two DNA-based diagnostic kits were reported to meet the requirements of in vitro diagnostic directives for the optimal diagnosis of STDs, Aptima Combo yielded false positive results of CT and NG [20].

Le Roy et al. [21] investigated the diagnostic performance of a BioRad Dx CT/NG/MG assay for simultaneous detection of CT, NG, and MG using 453 clinical samples. The authors reported the prevalence of CT infection among men and women with urogenital symptoms as 8.3% (19/236) and 11.1% (3/27), respectively. The prevalence of MG infection was 8.3% (2/24) and 3.7% (1/27) in symptomatic men and women, respectively [21]. The clinical sensitivities of the kit in the diagnosis of CT and MG infections were calculated as 100% for both men and women, irrespective of the presence of any symptom. Of seven NG-positive samples, all were confirmed by the culture method. Co-infections of CT with NG and CT with MG were detected in two and three patients, respectively. The prevalence of CT (11.1%) and MG (7.8%) among men detected in our study, which included patients with urogenital symptoms, was parallel to the findings of the study carried out by Le Roy et al. [21]. The prevalence of CT, MG, and NG among women found in our study (11.1%, 7.8%, and 3.3%, respectively) was slightly higher, and the prevalence of NG among men (4.4%) was approximately four-fold lower than that reported [21]. Consistent with the findings of Le Roy et al. [21], single infections of CT, MG, and NG were detected in 28, 20, and 7 patients, whereas dual infections of CT–MG, NG–CT, and NG–MG were reported in six, five, and one patient(s), respectively. In addition to dual infections, a triple co-infection with CT, MG, and NG was detected in one patient.

CT and NG are the causative agents in the overwhelming majority of cervicitis. However, the role of MG and *Ureaplasma* spp. in the etiology of cervicitis and the association of the bacterial load with the disease still need to be better understood. In a study carried out in North India that investigated the prevalence of CT, NG, MG, and *Ureaplasma* species and the association of their loads with cervicitis using the quantitative real-time PCR method, *U. parvum*, MG, NG, and CT were detected in 47 (31.3%), 37 (24.6%), 10 (6.6%), and 5 (3.3%) of 150 patients, respectively. Bacterial load was not associated with cervicitis, whereas old age and irregularity in menstrual cycles were found to be associated with cervicitis. MG was the only bacterial pathogen that was found to be associated with cervicitis according to age, presence of discharge, and dysuria [22]. In contrast to the study carried out by Roy et al. [22], in our study CT was the most frequently (21.4%) detected bacterial pathogen in cases with cervicitis, followed by MG and NG (3.1% each). Differences in the prevalence of CT, MG, and NG infections between the two studies may be attributed to geographical differences and demographical variations of the patients included. Moreover, our study did not detect MG in patients older than 41 years of age. The contradiction between our finding and the assumption by Roy et al. [22] may arise due to a limited number of elderly patients included in our study.

De Souza et al. [23] investigated the presence of sexually transmitted bacteria in urethral swabs of 170 men by using conventional methods and molecular tests. The authors detected NG in 102 (60.0%), CT in 50 (29.4%), *U. urealyticum* in 29 (17.0%), MG in 11 (6.5%), *U. parvum* in 10 (5.9%), and *M. hominis* in 7 (4.1%) patients by the PCR method. Single infection with NG, CT, and MG was reported in 46 (27.1%), 13 (7.6%), and 5 (2.9%) patients, respectively. The co-infection rate was 40.6%, and the most frequently detected co-infection was dual infection with NG and CT. As a result of the study, it was concluded that the rate of co-infection is high and molecular techniques are advantageous for the rapid and accurate diagnosis of the infections, which are crucial for both the treatment and prevention of STDs [23]. In 138 patients with urethritis included in our study, NG (5.8%), CT (13%), and MG (10.1%) were detected at higher rates, and the overall co-infection rate (4.1%) was approximately tenfold lower than that reported by de Souza et al. [23].

STDs are among the important causes of vaginitis in women. Therefore, women with vaginitis should be tested for sexually transmitted pathogens. Recently, molecular tests that can be used to determine vaginitis’s etiological agent have provided broad diagnostic coverage of pathogens, including several bacteria and the parasite *T. vaginalis*. In a study carried out in Canada, including 560 women with vaginitis, *T. vaginalis* was detected in 69 (12.3%), CT in 34 (6.1%), and NG in 10 (1.8%) patients. Women who had bacterial vaginosis or concurrent infection with *Candida* spp. were reported to be at higher risk (24.4 and 25.7%, respectively) of coinfection with sexually transmitted pathogens. Moreover, women without vaginitis were determined to have STDs at a lower rate (7.9%) than the control group [24]. Similarly, 11 (7.9%) and 3 (2.1%) of the women with vaginitis included in our study were detected to be CT- and NG-positive, respectively.

STDs are generally more common among infertile individuals than those who are fertile. In a study by Tomusiak et al. [25] in Poland, *U. urealyticum*, *M. hominis*, and CT were identified in 9% and 8%, 4% and 0%, and 0% and 3% of 101 infertile and 60 fertile women, respectively. In various studies, infection with CT among women was shown to be extensively associated with infertility [26,27]. Paira et al. [28] reported that CT, *Ureaplasma* spp., and *M. hominis* positivities were higher in infertile patients than in fertile patients (5.3%, 22.8%, and 7.4% vs. 2%, 17.8%, and 1.7%, respectively). CT and *M. hominis* were dominant agents in infertile men, whereas *Ureaplasma* spp. and *M. hominis* were more prevalent in women with infertility. As a result, the authors recommended the screening of urogenital infections among patients with infertility [28]. Of the 30 infertile patients included in our study, 2 were found to be CT-positive.

Because of similar transmission routes, individuals with any STDs are at higher risk for acquiring HIV infection and other STDs. Therefore, HIV and other STD tests should be offered to individuals diagnosed with or at risk for a particular STD. Genital ulcers and non-ulcerative STDs are also known to facilitate the transmission of HIV because of disrupting mucosal barriers. Because STDs are biological markers of risk for HIV acquisition and transmission, STD screening is crucial as a component of STD and HIV risk assessment among individuals at risk of HIV infection. On the other hand, in individuals living with HIV, STDs may present with atypical signs and can be persistent or recurrent [7], [29]. In our study, single infections of MG and CT were detected in a woman and a man living with HIV. Co-infection of NG with CT was also identified in a woman living with HIV.

Socioeconomic variations are known to affect the prevalence of STDs. Individuals living in rural areas are at higher risk of acquiring STDs than those in urban regions because of a lack of awareness. However, a recent increase in migration from rural to urban areas and a rise in the population size of the urban regions tend to increase the prevalence of STDs in these areas [30,31]. Although statistically insignificant in our study, bacteria responsible for STDs were more frequently detected in individuals living in urban regions than those in suburban regions. These data can be attributed to the higher number of people inhabiting urbanized regions, increased sexual partnership concurrency, other high-risk behaviors, and socialization patterns in these areas.

The FTD Urethritis basic kit used in this study is an in vitro diagnostics (IVD) qPCR-based commercially available kit. This kit has very high sensitivity and specificity and is considered the gold standard method in different studies. The test results are 100% reproducible, supporting the validity of the kit’s results [32]. Nucleic acid amplification technologies (NAATs) based on commercially available kit sensitivity are thought to be better than other detection methods [33]. Harrison et al. reported the sensitivity, specificity, PPV, and NPV values of the FTD Urethritis kit in urogenital specimens as 100%, 99.6%, 96%, and 100%, respectively [34]. After calculation with the Rogan–Gladen estimator; for CT, Apparent Prevalence (WilsonCL) and True Prevalence (BlakerCL) were found to be 0.1111 (11.11%) and 0.1021 (10.21%), respectively. For NG, apparent prevalence (WilsonCL) and true prevalence (BlakerCL) were found to be 0.0389 (3.89%), and 0.0292 (2.92%), respectively. For MG, apparent prevalence (WilsonCL) and true prevalence (BlakerCL) were found to be 0.0778 (7.78%) and 0.0685 (6.85%), respectively.

A comparative table summarizing the distribution of the prevalence data of different studies is presented (Table 5). It was found that differences could be detected even in the same country data and that the prevalence values of targets such as CT, NG, and MG may vary according to the method, sample, and clinical situation. When overall prevalence rates were analyzed, it is seen that the prevalence of CT is higher than NG and MG, except for the study in Brazil [23]. In terms of the relationship between CT and infertility, the incidence of CT was found to be 4% and above in other studies and in our study. Regarding the relationship between STD agents and specific diseases such as cervicitis and urethritis, it has been found that different prevalence rates were found in different studies.

The limitation of this study is that it was single-centered, and only a commercially available real-time PCR-based multiplex kit was used to obtain data on targets. Therefore, our study did not use culture or microscopy methods to detect targets. However, these data are valuable, as the sensitivity and specificity of the kit in urogenital specimens are pretty high. This study also showed that the prevalence data might vary even in the same country and that multiplex real-time PCR-based kits can be used safely in urogenital samples and detect co-infections.

## 5. Conclusions

In this study, among *C. trachomatis, N. gonorrhoeae,* and *M. genitalium*, CT was found to be the most frequently detected bacterial agent of STDs in our hospital at Istanbul. Co-infections that comprise more than one-fifth of the cases should not be underestimated. In addition to raising community awareness about sexual and reproductive health so that sexual behaviors that place individuals at risk of acquiring STDs are avoided, regular screening of the individuals and follow-up of the causative agents using multiplex real-time PCR-based diagnostic methods that enable immediate detection of co-infections are essential for the treatment and the primary prevention of STDs. Due to the health hazards and social and economic losses caused by STDs, different STD prevention strategies, such as routine monitoring of prevalence data and the introduction of rapid diagnostic tests, should be implemented to reduce the risk of contracting and transmitting diseases.

## Figures and Tables

**Figure 1 healthcare-11-00930-f001:**
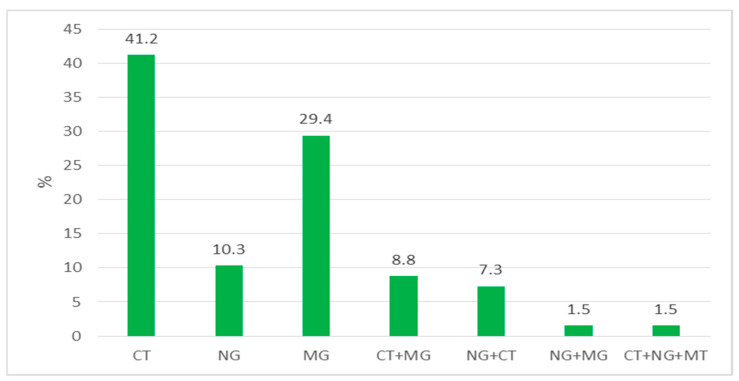
Distribution of the single infections and co-infections among infected patients.

**Table 1 healthcare-11-00930-t001:** Distribution of sexually transmitted bacteria according to the clinical presentation (CP), gender, age, and settlement of patients.

		Total [*n* (%)]	CT [*n* (%)]	NG [*n* (%)]	MG [*n* (%)]	p1 * [*p* (95%CI)] **	p2 * [*p* (95%CI)] **	p3 * [*p* (95%CI)] **
Overall		360	40 (11.1)	14 (3.9)	28 (7.8)			
CP	Urethritis	138 (38.3)	18 (13.0)	8 (5.8)	14 (10.1)	0.052	0.233	0.323
	Vaginitis ***	140 (38.9)	11 (7.9)	3 (2.1)	11 (7.9)			
	Cervicitis ***	42 (11.7)	9 (21.4)	3 (7.1)	3 (7.1)			
	Infertility	30 (8.3)	2 (6.7)	-	-			
	Others ^Ø^	12 (2.8)	-	-	-			
Gender #	M	180 (50)	20 (11.1)	8 (4.4)	14 (7.8)	-	0.586 [1.35 (0.46–3.97)]	-
	F	180 (50)	20 (11.1)	6 (3.3)	14 (7.8)			
Age (years)	≤20	8 (2.2)	-	-	-	0.145	0.455	0.002
	21–30	136 (37.8)	21 (15.4)	8 (5.9)	19 (14.0)			
	31–40	110 (30.5)	13 (11.8)	3 (2.7)	8 (7.2)			
	41–50	70 (19.4)	4 (5.7)	3 (4.3)	-			
	≥51	36 (10)	2 (5.5)	-	-			
Location	Urban	320	38 (11.9)	14 (4.4)	27 (8.4)	0.192 [0.4 (0.1–0.7)]	0.177 [0.96 (0.93–0.98)]	0.313 [0.28 (0.04–2.1)]
	Suburban	40	2 (5)	-	1 (2.5)			

* p1: CT, p2: NG, p3: MG; ** C.I.: confidence interval; *** dual diagnosis; Ø salpingitis, epididymitis and PID, # M: male, F: female.

**Table 2 healthcare-11-00930-t002:** Distribution of single infections and co-infections according to the age and gender of the infected patients (*n*).

Age	Gender		Single Infection/Co-Infection						
	Female	Male	CT	NG	MG	CT + MG	NG + CT	NG + MG	CT + NG + MG
≤20	1	-	-	-	1	-	-	-	-
21–30	18	19	12	3	12	5	3	1	1
31–40	10	12	11	2	7	1	1	-	-
41–50	3	3	3	2	-	-	1	-	-
≥51	1	1	2	-	-	-	-	-	-
Total	33	35	28	7	20	6	5	1	1

**Table 3 healthcare-11-00930-t003:** Distribution of co-infection status with different STI agents of these patients (*n*).

Agents	Gender		Single Infection/Co-Infection						
	Female	Male	CT	NG	MG	CT + MG	NG + CT	NG + MG	CT + NG + MG
HIV (n: 3)	1	2	1	-	1	-	1	-	-
*Treponema pallidum* (n: 3)	1	2	1	1	-	-	1	-	-
*Trichomonas vaginalis* (n: 2)	1	-	1	-	-	-	-	-	-
HPV (n: 2)	-	2	1	1	-	-	-	-	-
HBV (n: 1)	1	-	-	-	1	-	-	-	-
HSV2 (n: 1)	1	-	1	-	-	-	-	-	-

**Table 4 healthcare-11-00930-t004:** Estimated true prevalence in this study with Rogan–Gladen estimator.

Target		Estimate	Lower 95% CL	Upper 95% CL
CT	Apparent Prevalence (WilsonCL)	0.1111	0.0827	0.1478
	True Prevalence (BlakerCL)	0.1021	0.0734	0.1392
NG	Apparent Prevalence (WilsonCL)	0.0389	0.0233	0.0642
	True Prevalence (BlakerCL)	0.0292	0.0134	0.0548
MG	Apparent Prevalence (WilsonCL)	0.0778	0.0544	0.1101
	True Prevalence (BlakerCL)	0.0685	0.0448	0.1011

**Table 5 healthcare-11-00930-t005:** Distribution of prevalence data of different studies.

		CT [*n* (%)]	NG [*n* (%)]	MG [*n* (%)]
In this study	Istanbul/Turkey	40 (11.1)	14 (3.9)	28 (7.8)
De Souza et al. [23]	Brazil	50 (29.4)	102 (60.0)	11 (6.5)
Van Der Pol et al. [24]	Canada	34 (6.1)	10 (1.8)	
Le Roy et al. [21]	France	(8.3)		(5.8)
Calas et al. [35]	Reunion Island	(10.2)	(1.07)	(2.69)
Koksal et al. [36]	Istanbul/Turkey	57 (13.6)		
Guralp et al. [37]	Cyprus	(5.4)	(2.5)	(2.9)
In this study	Infertility	2 (6.7)	-	-
Tomusiak et al. [25]	Infertility	(4)		
Paira et al. [28]	Infertility	(5.3)		
In this study	Cervicitis	9 (21.4)	3 (7.1)	3 (7.1)
Roy et al. [22]	Cervicitis	5 (3.3)	10 (6.6)	37 (24.6)
In this study	Urethritis	18 (13.0)	8 (5.8)	14 (10.1)
De Souza et al. [23]	Urethritis	13 (7.6)	46 (27.1)	5 (2.9)

## Data Availability

No new data were created.

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
