# Peer review of "Prevalence of Chlamydia trachomatis, Neisseria gonorrhoeae, and Mycoplasma genitalium among Patients with Urogenital Symptoms in Istanbul"

_healthcare, 2023, doi:10.3390/healthcare11070930_

Round 1

Reviewer 1 Report (Previous Reviewer 2)

The paper has improved from the last version.

Author Response

Thank you very much for your valuable comments and contribution. all comments have been carefully checked and edited on the manuscript.

You can also see our response below.

Thank you, Regards

Assoc. Prof. Hayriye Kirkoyun Uysal

Response to Reviewer 1:

The paper has improved from the last version.

Response: Thank you very much for your valuable comments and contribution.

Reviewer 2 Report (Previous Reviewer 3)

The authors report that the study was planned as randomized study. This formulation is unclear and should be elaborated in detail (what exactly was planned to randomize and was the planning also implemented?).

Author Response

Thank you very much for your valuable comments and contribution. all comments have been carefully checked and edited on the manuscript.

You can also see our response below.

Thank you, Regards

Assoc. Prof. Hayriye Kirkoyun Uysal

Response to Reviewer 2:

The authors report that the study was planned as randomized study. This formulation is unclear and should be elaborated in detail (what exactly was planned to randomize and was the planning also implemented?).

Response: Thank you very much for your valuable comments and contribution. Although our study was planned as a complete randomisation study, we did not want to focus on only one gender and therefore proceeded with our study in the block randomisation-based method. In line with your suggestion, we stated in the manuscript that we carried out a study with block randomisation method.

As you know, A key advantage of blocked randomization is that treatment groups will be equal in size and will tend to be uniformly distributed by key outcome-related characteristics. Complete randomization can however create imbalanced designs, for example, grouping all samples of the same condition in the same batch. Block randomization is an approach that can prevent severe imbalances in sample allocation with respect to both known and unknown confounders.

Efird J. Blocked randomization with randomly selected block sizes. Int J Environ Res Public Health. 2011;8(1):15-20. doi:10.3390/ijerph8010015

Nair B. Clinical Trial Designs. Indian Dermatol Online J. 2019;10(2):193-201. doi:10.4103/idoj.IDOJ_475_18

This manuscript is a resubmission of an earlier submission. The following is a list of the peer review reports and author responses from that submission.

Round 1

Reviewer 1 Report

     The subject of the work is of interest. However, there are some points that need consideration:

1)      There is no clear answer to the research question from this study.

2)      Authors did not comment on the gap of knowledge the current research will try to fill.

3)      Report the ethical code.

4)     It is recommended to discuss the prevalence of these infections in the form of a diagram based on previous studies.

Author Response

Thank you very much for your valuable comments and contribution. all comments have been carefully checked and edited on the manuscript.

You can also see our response below.

Thank you, Regards

Assoc. Prof. Hayriye Kirkoyun Uysal

Response to Reviewer 1:

The subject of the work is of interest. However, there are some points that need consideration:

1)      There is no clear answer to the research question from this study.

Response: Thank you very much for your valuable comments and contribution. on your suggestion, the conclusion part has been revised.

2)      Authors did not comment on the gap of knowledge the current research will try to fill.

Response: Thank you very much for your valuable comments and contribution. on your suggestion, the importance of the results of our research has been explained.

 3)      Report the ethical code.

Response: Thank you very much for your valuable comments and contribution. Ethical committee approval number and approval date were added to the method section.

4)     It is recommended to discuss the prevalence of these infections in the form of a diagram based on previous studies.

Response: Thank you very much for your valuable comments and contribution. Upon your important suggestion, the discussion has been expanded and the summary table and prevalence data have been added to the discussion section.

Reviewer 2 Report

The aurthors aimed to detect and determine the prevalence of Chlamydia trachomatis, Neisseria gonorrhoeae, and Mycoplasma genitalium  in Turkish patients who had urogenital symptoms. A total of 360 samples was used for the analysis with 180 being female or male. They used a multiplex pcr to detect pathogens.  They found 68 patients having STIs.  Younger patients 21-30 years represented one third of the population. They show a prevalence of 11%, 3.9%, 7.8% of C. trachomatis, N. gonorrhoeae, and M. genitalium. The primary pathogen in the urogenital samples of men was CT (11.1%) followed by MG (7.8%) and, NG (4.4%). CT (11.1%) was the most frequently detected pathogen in the samples collected from women followed by MG (7.8%), and NG (3.3%). the author suggest that routine screening should be performed.

The aurthor's should also include all other methods that was used to diagnose STI such as culture or even if microscopy was performed on specimens. 

They should include more detail on the methods and kits used.

Throughout manuscript the sentences a have few refences, more should be included.

Author Response

Thank you very much for your valuable comments and contribution. all comments have been carefully checked and edited on the manuscript.

You can also see our response below.

Thank you, Regards

Assoc. Prof. Hayriye Kirkoyun Uysal

Response to Reviewer 2:

The authors aimed to detect and determine the prevalence of Chlamydia trachomatis, Neisseria gonorrhoeae, and Mycoplasma genitalium  in Turkish patients who had urogenital symptoms. A total of 360 samples was used for the analysis with 180 being female or male. They used a multiplex pcr to detect pathogens.  They found 68 patients having STIs.  Younger patients 21-30 years represented one third of the population. They show a prevalence of 11%, 3.9%, 7.8% of C. trachomatis, N. gonorrhoeae, and M. genitalium. The primary pathogen in the urogenital samples of men was CT (11.1%) followed by MG (7.8%) and, NG (4.4%). CT (11.1%) was the most frequently detected pathogen in the samples collected from women followed by MG (7.8%), and NG (3.3%). the author suggest that routine screening should be performed.

Response: Thank you very much for your valuable comments and contribution.

The aurthor's should also include all other methods that was used to diagnose STI such as culture or even if microscopy was performed on specimens.

Response: Thank you very much for your valuable comments and contribution. Culture or microscopy methods were not used to detect targets in our study. But, FTD Urethritis basic kit used in this study is an in vitro diagnostics (IVD) qPCR-based commercially available kit. This kit has very high sensitivity and specificity and is considered the gold standard method in different studies. The test results are 100% reproducible, supporting the validity of the kit's results. Nucleic acid amplification technologies (NAATs) based commercially available kit sensitivity is thought to be better than other detection methods.

Regarding this situation you mentioned, the limitation section of our study has been added and sentences have been added to the discussion of the sensitivity, specificity, PPV and NPV values of the kit used.

They should include more detail on the methods and kits used.

Response: Thank you very much for your valuable comments and contribution. More information about the method has been added.

Throughout manuscript the sentences a have few refences, more should be included.

Response: Thank you very much for your valuable comments and contribution. More refences has been added.

Reviewer 3 Report

Overall, a well-written paper. Unfortunately, the diagnostic accuracy of the PCR used (sensitivity, specificity) were not described. This should be added.  Subsequently, the prevalence estimates should be adjusted for the diagnostic accuracy of the PCR used (Rogan-Gladen estimator). Positive and negative predictive values should be added as well. In the discussion, the diagnostic accuracy should be addressed with regard to the reliability of the results found. 

Author Response

Thank you very much for your valuable comments and contribution. all comments have been carefully checked and edited on the manuscript.

You can also see our response below.

Thank you, Regards

Assoc. Prof. Hayriye Kirkoyun Uysal

Response to Reviewer 3:

Overall, a well-written paper. Unfortunately, the diagnostic accuracy of the PCR used (sensitivity, specificity) were not described. This should be added.  Subsequently, the prevalence estimates should be adjusted for the diagnostic accuracy of the PCR used (Rogan-Gladen estimator). Positive and negative predictive values should be added as well. In the discussion, the diagnostic accuracy should be addressed with regard to the reliability of the results found. 

Response: Thank you very much for your valuable comments and contribution. FTD Urethritis basic kit used in this study is an in vitro diagnostics (IVD) qPCR-based commercially available kit and this kit can use as a gold standart in different studies. Sentences have been added to the discussion of the specificity, sensitivity, PPV and NPV data for this kit, as you indicated. Since the sensitivity and specificity of the kit were found to be close to 100% in studies with urogenital samples, True prevalence data were not recalculated with the Rogan-Gladen estimator.

Round 2

Reviewer 3 Report

Unfortunately, my comments were not sufficiently addressed. The evaluation of PCR addressed by the authors in the discussion does not refer to the PCR used in the study. In addition, the information referenced there is contradictory (on the one hand, perfect specificity is given for Aptima, on the other hand, reference is made to false-positive results). The reported specificity for the second PCR evaluated (CoBAS) is also rather low at 96.67%. Another evaluation for the Bio-Rad Dx CT/NG/MG assay reports higher sensitivities and specificities. However, especially with low prevalences, even slightly reduced specificities can lead to a significant overestimation of the prevalence. For publication, the diagnostic accuracy of the PCR used must be presented and discussed in detail. The prevalence must be adjusted for the diagnostic accuracy. Otherwise, the reported prevalence is only the test positive rate and should be described as such (i.e. not misleadingly as prevalence).

Author Response

Dear Reviewer,

Thank you very much for your valuable comments and contribution. all comments have been carefully checked and edited on the manuscript.

You can also see our response below.

Thank you, Regards

Assoc. Prof. Hayriye Kirkoyun Uysal

Response to Reviewer 3:

Round 1

Overall, a well-written paper. Unfortunately, the diagnostic accuracy of the PCR used (sensitivity, specificity) were not described. This should be added.  Subsequently, the prevalence estimates should be adjusted for the diagnostic accuracy of the PCR used (Rogan-Gladen estimator). Positive and negative predictive values should be added as well. In the discussion, the diagnostic accuracy should be addressed with regard to the reliability of the results found. 

Response: Thank you very much for your valuable comments and contribution. FTD Urethritis basic kit used in this study is an in vitro diagnostics (IVD) qPCR-based commercially available kit and this kit can use as a gold standart in different studies. Sentences have been added to the discussion of the specificity, sensitivity, PPV and NPV data for this kit, as you indicated. Since the sensitivity and specificity of the kit were found to be close to 100% in studies with urogenital samples, True prevalence data were not recalculated with the Rogan-Gladen estimator.

Round 2

Unfortunately, my comments were not sufficiently addressed. The evaluation of PCR addressed by the authors in the discussion does not refer to the PCR used in the study. In addition, the information referenced there is contradictory (on the one hand, perfect specificity is given for Aptima, on the other hand, reference is made to false-positive results). The reported specificity for the second PCR evaluated (CoBAS) is also rather low at 96.67%. Another evaluation for the Bio-Rad Dx CT/NG/MG assay reports higher sensitivities and specificities. However, especially with low prevalences, even slightly reduced specificities can lead to a significant overestimation of the prevalence. For publication, the diagnostic accuracy of the PCR used must be presented and discussed in detail. The prevalence must be adjusted for the diagnostic accuracy. Otherwise, the reported prevalence is only the test positive rate and should be described as such (i.e. not misleadingly as prevalence).

Response: Thank you very much for your valuable comments and contribution. We apologise for the misunderstanding. The multiplex real-time PCR kit used in the study was referred to in the following sentence. The kit of Fast Track Diagnostics was used in our study and the sensitivity, specificity, PPV and NPV values of this kit were referred to in the discussion and the questions raised by the referee were tried to answer. "Harrison et al. reported the sensitivity, specificity, PPV, and NPV values of the FTD Urethritis kit in urogenital specimens as 100%, 99.6%, 96%, and 100%, respectively [34]."

In line with the recommendations of the reviewer, the true prevalence was determined by calculating the Rogan-Gladen estimator taking these data into account.

True Prevalence=  Apparent Prevalence + (Specificity − 1) / Specificity + (Sensitivity − 1) formula was used for the Rogan-Gladen estimator.

After calculation with the Rogan-Gladen estimator; for CT, Apparent Prevalence (WilsonCL) and True Prevalence (BlakerCL) were found 0.1111 (11.11%) and 0.1021 (10.21%) respectively. for NG, Apparent Prevalence (WilsonCL) and True Prevalence (BlakerCL) were found 0.0389 (3.89%), and 0.0292 (2.92%) respectively. for MG, Apparent Prevalence (WilsonCL) and True Prevalence (BlakerCL) were found 0.0778 (7.78%) and 0.0685 (6.85%).

Calculations of the true prevalence were also included in the manuscript.
